

# A serendipity-biased Deepwalk for collaborators recommendation

Zhenzhen Xu, Yuyuan Yuan, Haoran Wei and Liangtian Wan

Key Laboratory for Ubiquitous Network and Service Software of Liaoning Province, School of Software, Dalian University of Technology, Dalian, Liaoning, China

## ABSTRACT

Scientific collaboration has become a common behaviour in academia. Various recommendation strategies have been designed to provide relevant collaborators for the target scholars. However, scholars are no longer satisfied with the acquainted collaborator recommendations, which may narrow their horizons. Serendipity in the recommender system has attracted increasing attention from researchers in recent years. Serendipity traditionally denotes the faculty of making surprising discoveries. The unexpected and valuable scientific discoveries in science such as X-rays and penicillin may be attributed to serendipity. In this paper, we design a novel recommender system to provide serendipitous scientific collaborators, which learns the serendipity-biased vector representation of each node in the co-author network. We first introduce the definition of serendipitous collaborators from three components of serendipity: relevance, unexpectedness, and value, respectively. Then we improve the transition probability of random walk in DeepWalk, and propose a serendipity-biased DeepWalk, called Seren2vec. The walker jumps to the next neighbor node with the proportional probability of edge weight in the co-author network. Meanwhile, the edge weight is determined by the three indices in definition. Finally, Top-N serendipitous collaborators are generated based on the cosine similarity between scholar vectors. We conducted extensive experiments on the DBLP data set to validate our recommendation performance, and the evaluations from serendipity-based metrics show that Seren2vec outperforms other baseline methods without much loss of recommendation accuracy.

## INTRODUCTION

In academia, the rapid accumulation of scholarly data has produced an overload of academic information, and scholars are lost in it because of the difficulty in accessing useful information. The appearance of a recommender system alleviates the problem, i.e., providing relevant collaborators for target scholars, which focuses on improving the recommendation accuracy. Most recommendation approaches build the profiles of target scholars based on their interests or research contents, and then provides a list of collaborators who have similar profiles with them.

However, researchers are no longer satisfied with the relevant or acquainted recommendations, which may narrow their horizons in the long term. Furthermore,

Corresponding author
Liangtian Wan,
wan.liangtian.2015@ieee.org

accuracy is not the absolute metric in determining good recommendation performance, and it sometimes hurts the recommender systems for lacking novelty and diversity (*Sean, McNee & Konstan, 2006*; *McNee, Riedl & Konstan, 2006*). Under this circumstance, serendipity is taken into consideration in terms of satisfying users when designing or evaluating the recommender systems (*Kotkov, Wang & Veijalainen, 2016*). The concept of serendipity can be understood flexibly in most cases, which has different implications under different scenarios. Additionally, the serendipitous encounters are rare in both academia and daily life of researchers. Therefore, no consensus is reached on the definition of serendipity.

In this paper, we aim to recommend the serendipitous scientific collaborators for target scholars by learning the vector representation of each scholar node in co-author network. The first essential step of this work is the definition of serendipitous collaborators. We induce the definition by three components of serendipity, which are relevance, unexpectedness, and value, respectively. Relevance is quantified as the proximity between two connected nodes in co-author network. Unexpected collaborators have research topics that are different from the topics of their target scholars; therefore, they usually have diverse research topics compared with their target scholars. While the value of a collaborator is quantified as the eigenvector (*Bonacich, 2007*) of the collaborator node in the co-author network, which represents the influence of this collaborator. According to the nature of serendipity, we define that serendipitous collaborators are more unexpected and valuable, but less relevant for their target scholars. We reserve lower relevance for the significance of serendipity, because relevance caters to the preferences of target scholars. While the core components of serendipity are unexpectedness and value. Consequently, the intuitive definition is that a serendipitous collaborator has high topic diversity, high influence and low network proximity relatively for his/her target scholar. The second essential step is the design of appropriate recommendation algorithm. Though collaborative filtering is the most universal recommendation approach (*Kim et al., 2010*; *Konstas, Stathopoulos & Jose, 2009*), it is not yet applicable to our recommendation scenario. Recently, researchers have shown increasing interest in the technology of network embedding. The vector representations of network nodes have also been applied to many prediction and recommendation tasks by learning relevant features of nodes successfully (*Perozzi, Al-Rfou & Skiena, 2014*; *Grover & Leskovec, 2016*; *Tian & Zhuo, 2017*). In this case, we design a serendipitous collaborators recommendation strategy by improving DeepWalk (*Perozzi, Al-Rfou & Skiena, 2014*), where the walker jumps to the next node based on the proportional probability of its edge weight with the connected node in co-author network. Besides, the edge weight between a collaborator and his/her target scholar is determined by the extent of serendipity. Therefore, this is a serendipity-biased DeepWalk for learning the vector representation of each author node in co-author network, and we call it Seren2vec.

Seren2vec embeds each author node with a low-dimensional vector that carries serendipity between collaborators in the co-author network. We extract Top-N collaborators for recommendation based on the cosine similarity between author vectors. To the best of our knowledge, this is the first work that takes serendipity into consideration when designing the collaborators recommender system. Our strategy enabled us to simultaneously improve the serendipity of the recommendation list and to maintain

adequate accuracy. The definition of serendipitous collaborators is also significant for further mining the interesting collaboration mechanism in science. The main contributions of this paper can be summarized as follows:

- Define serendipitous scientific collaborators: we propose the intuitive definition of serendipitous scientific collaborators from three indices, which are network proximity, topic diversity, and collaborator influence, respectively.
- Propose a serendipity-biased DeepWalk (Seren2vec): we improve the process of random walk in DeepWalk for serendipitous collaborators recommendation. The walker jumps to the next neighbor node with the proportional probability of its edge weight with the connected node, and the edge weight is determined by the extent of serendipity. Therefore, the vector representation of each author node is serendipity-biased.
- Recommend serendipitous scientific collaborators: we perform Seren2vec to learn the representation of each author node with low-dimensional vector, and then extract the Top-N similar collaborators by computing the cosine similarity between the target vector and other author vectors.
- Analyze and evaluate the recommendation results: we conduct extensive experiments on the subset of DBLP, and evaluate the recommendation results from both accuracy-based and serendipity-based metrics. Furthermore, we compare the recommendation performance of Seren2vec with other baseline methods for validating our scheme.

In the following sections: we first briefly review the related work, including the widely used collaborators recommendation approaches and the integration of serendipity in recommender systems. The proposed definition of serendipitous scientific collaborators and the corresponding indices are analyzed and quantified in 'The Definition of Serendipitous Collaborators'. The framework of our method is discussed in 'The Framework of Seren2vec'. The experimental results and metrics for evaluation are presented in 'Experiments'. Finally we conclude the paper in the last section.

## RELATED WORK

Researchers have designed various academic recommender systems for scientific applications. However, most existing recommendation approaches aim to improve the recommendation accuracy based on the profile similarity between users. Long-term use of such recommender systems will degrade the user satisfaction, since most recommendations are the acquaintances of target users. The serendipity-related elements have attracted increasing attention from researchers for designing serendipitous recommender systems, such as novelty, unexpectedness, diversity, etc. In this section, we summarize the widely used collaborators recommendation approaches, the existing technologies for integrating serendipity into recommendation systems, and the serendipity-based metrics for evaluating the recommendation results.

## Scientific collaborators recommender systems

Methods for recommending scientific collaborators have been studied for decades. The recommendation methods can be divided into content-based, collaborative filtering-based, and random walk-based algorithms on the whole.

### Content-based recommendation

Content-based method is a basic and widely used technique for recommending collaborators. The critical process of content-based collaborators recommendation is the scholar profile modelling, where the interests or topics of scholars can be inferred from their publication contents by extracting the words from title, abstract, keywords, etc. Many topic modeling techniques such as Word2vec (*Goldberg & Levy, 2014*), LDA (Latent Dirichlet Allocation) (*Blei, Ng & Jordan, 2003*) and pLSA (Probabilistic Latent Semantic Analysis) (*Hofmann, 2017*) enable to generate the probability distribution of these words, and they also contribute to generate the feature descriptions of different scholars. A collaborator recommendation list is finally generated by computing the profile similarity between scholars. *Gollapalli, Mitra & Giles (2012)* proposed a recommendation model by computing the similarity between expertise profiles. Besides, the expertise profiles are extracted from researchers' publications and academic home pages. *Lopes et al. (2010)* took the area of researchers' publications and the vector space model to make collaboration recommendation. The scientific collaboration mechanism is complex for the various factors. *Wang et al. (2017)* investigated the academic-age-aware collaboration behaviors, which may guide and inspire the collaborators recommendation strategies.

### Collaborative filtering-based recommendation

Collaborative filtering-based method is popular in the field of recommender system. The core of collaborative filtering is to find items via the opinions of other similar users whose previous history strongly correlates with the target user. In other words, the similar users have similar interests with the target user (*Jannach et al., 2010*). Finally, the items positively rated by the similar users will be recommended to the target user. *Heck, Peters & Stock (2011)* performed a collaborative filtering method via the co-citation and bibliographic coupling to detect author similarity. However, the cold start problem exists to degrade the recommendation performance, because it is difficult to find similar scholars without enough information of a new scholar. Content-based algorithms require contents to analyze by utilizing Natural Language Processing (NLP) tools, therefore the collaborative filtering-based algorithms are more easy to implement without the requirement of contents (*Hameed, Al Jadaan & Ramachandram, 2012*).

### Random walk-based recommendation

Random walk is the most common technique for collaborators recommendation. The basic idea of Random walk is to compute the structure similarity between nodes in co-author network based on the transition probability. *Xia et al. (2014)* explored three academic factors, i.e., coauthor order, latest collaboration time, and times of collaboration, to quantify the link importance and performed a biased random walk in academic social network for recommending most valuable collaborators. *Araki et al. (2017)* made use of

both topic model and random walk with restart method for recommending interdisciplinary collaborators. They combined the content-based and random-walk based methods for collaborators recommendation. Kong's works (*Kong et al., 2017*; *Kong et al., 2016*) exploited the dynamic research interests, academic influences and publication contents of scholars for collaborators recommendation. Random Walk can be improved flexibly by adjusting its transition matrix, therefore it have been widely used by researchers in their recommendation scenarios.

To sum up, all kinds of recommendation algorithms are designed based on the similarity between academic entities in order to enhance the recommendation accuracy. Most of them ignore the integration of other serendipity-related elements for the design of recommender systems. Consequently, they are not applicable to our task of recommending serendipitous collaborators.

## Serendipity in recommender systems

Increasing researchers are interested in investigating serendipity in recommender systems, *Kotkov, Wang & Veijalainen (2016)* wrote a survey to summarize the serendipity problem in recommender systems, including the related concepts, the serendipitous recommendation technologies and metrics for evaluating the recommendation results. They also analyzed the challenges of serendipity in recommender systems (*Kotkov, Veijalainen & Wang, 2016*). *Herlocker et al. (2004)* considered that serendipitous recommendations aim to help users finding interesting and surprising items that they would not discover by themselves. While Shani and Gunawardana (*Shani & Gunawardana, 2011*) described that serendipity associates with users' positive emotions on the novel items. The serendipitous items usually are unexpected and useful simultaneously for a user (*Manca, Boratto & Carta, 2015*; *Iaquinta et al., 2010*). From these perspectives, serendipity emphasizes the unexpectedness and value components.

### Serendipitous recommendation approaches

Various serendipity-enhancement approaches have been proposed by researchers. *Zhang & Hurley (2008)* suggested to maximize the diversity of the recommendation list and kept adequate similarity to the user query for avoiding monotonous recommendations. *Zhang et al. (2012)* proposed a Full Auralist recommendation algorithm in order to balance the accuracy, diversity, novelty and serendipity of recommendations simultaneously. However, the Auralist algorithm is difficult to realize for its complexity. *Said et al. (2013)* proposed a new k-furthest neighbor (KFN) recommendation algorithm in order to recommend novel items by selecting items which are disliked by dissimilar users. *Onuma, Tong & Faloutsos (2009)* proposed a novel TANGENT algorithm to broaden user tastes, and they aim to recommend the items in a graph that not only are relevant to the target users, but also have connectivity to other groups.

### Evaluation of serendipitous recommender systems

In terms of the evaluation of the proposed serendipitous recommendation strategies, the widely used metrics are unexpectedness, usefulness and serendipity, where serendipity is the combination of unexpectedness and usefulness metrics. *Adamopoulos & Tuzhilin (2015)*

considered that an unexpected item is not contained in the set of expected items of the target user, and the usefulness of an item can be approximated by the items' ratings given by users. *Murakami, Mori & Orihara (2007)* and *Ge, Delgado-Battenfeld & Jannach (2010)* indicated that the unexpectedness metric is determined by the primitive recommendation methods which produce relevant recommendations, and the item not in primitive recommendation set is regarded as an unexpected item. The serendipity of the recommendation list is determined by the rate of both unexpected and useful items (*Zheng, Chan & Ip, 2015*; *Ge, Delgado-Battenfeld & Jannach, 2010*; *Murakami, Mori & Orihara, 2007*). Meanwhile, *Zheng, Chan & Ip (2015)* suggested that the usefulness of a recommended item depends on whether the target user selects or favors it.

These literatures shed some light on the development of serendipitous recommender systems in different fields. Our work also refer to them for extracting the core components of serendipity, and evaluating the serendipitous recommendations from the serendipity-based metrics.

## THE DEFINITION OF SERENDIPITOUS COLLABORATORS

"Serendipity" has been recognized as one of the most untranslatable words. The term serendipity origins from a fairy tale, "The Three Princes of Serendip" (*West, 1963*). The three princes were always making fortunate discoveries in their wandering adventures, and the accidental but valuable discoveries denote serendipity. Nevertheless, it is unclear what makes a collaborator serendipitous to his/her target scholar. In this paper, we define serendipitous scientific collaborators from three components: relevance, unexpectedness and value, corresponding with the indices of network proximity, topic diversity, and collaborator influence, respectively. We describe the details of these indices and their quantifications in the following three subsections.

### Relevance score

Relevance between target scholar and his/her collaborators may be quantified with their proximity in co-author network. Therefore, we perform Random Walk with Restart (RWR) (*Vargas & Castells, 2011*; *Tong, Faloutsos & Pan, 2008*) in the co-author network for computing the relevance score of all collaborator nodes for their target scholars. Finally, we get the relevance score between each pair of collaborators.

### Unexpectedness score

Unexpected scientific collaborators have connectivity to different research areas of their target scholars. Such unexpected collaborators may expand target scholars' horizons, since they have diverse research topics compared with their target scholars. We get each scholar's research areas by LDA (*Blei, Ng & Jordan, 2003*), and a collaborator who has connectivity to other areas different from that of his/her target scholar is regarded as a unexpected collaborator. LDA computes the cosine similarity between the topic probability distribution of each scholar and area, where the topic probability distribution of a scholar is parsed from his/her publication contents, and the topic probability distribution of an area is extracted from the proportion of each topic contained in this area. The areas

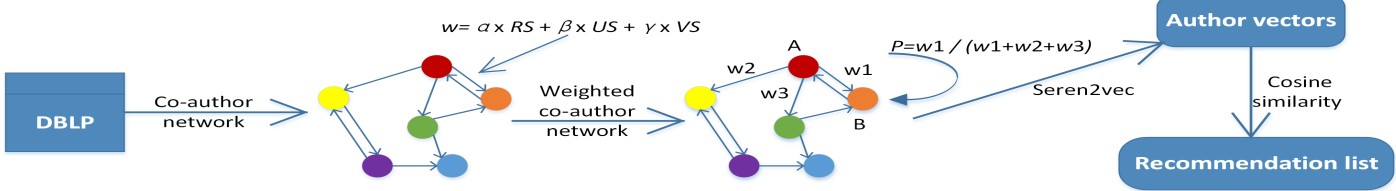

**Figure 1  The framework of Seren2vec.** The core of this framework is the attachment of serendipity to the collaboration edges, and the edge weight is quantified based on the three indices in definition in a linear way. Therefore, seren2vec contributes to a serendipity-biased representation learning.

with similarity higher than 0.6 are regarded as the research areas of a scholar. Finally we add the betweenness centrality (*Leydesdorff, 2007*) of a collaborator node with the number of communities it crosses as its unexpectedness score for its target scholar, since betweenness centrality represents the ability of a node to transfer information among separate communities in a network, which stresses the strong position of unexpected node.

## Value score

Value of a scholar is defined as the influence of this scholar node, and more influential nodes are more valuable for his/her collaborators in co-author network. We compute the value scores of all collaborator nodes with the Eigenvector Centrality (*Bonacich, 2007*), where the centralized value of a node is determined by the nodes linked by it. If the high-degree node connects with the target node, it usually are more valuable than those low-degree nodes for the target node. Besides, the influence of a node not only depends on its degree, but also depends on its neighbor nodes' influences according to the peculiarity of Eigenvector Centrality.

The relevance, unexpectedness, and value scores are computed between each pair of collaborators according to above quantification methods. A serendipitous collaborator has high unexpectedness score, high value score, and low relevance score for his/her target scholar relatively.

## THE FRAMEWORK OF SEREN2VEC

In this section, we propose the framework of Seren2vec, which aims to recommend serendipitous collaborators for scholars by learning the vector representation of each author node in co-author network. The whole framework of Seren2vec is shown in Fig. 1, which contains four main steps:

(1)  Compute the relevance, unexpectedness and value scores of each collaborator for his/her target scholar.
(2)  Construct a co-author network based on the collaboration data extracted from DBLP, where the edge weight is determined by the linear combination of relevance, unexpectedness and value scores.
(3)  Perform Seren2vec in co-author network, where the walker jumps to the next node *B* from node *A* with the proportional probability of their edge weight. The edge weight

is attached with the extent of serendipity, i.e., $w1$ in Fig. 1 shows the serendipity extent brought by $B$ for target scholar $A$.

(4)   Learn the representations of all author nodes with low dimensional vectors, and then compute the cosine similarity between author vectors for generating the recommendation list composed of Top-N similar collaborators.

## Integrate serendipity into co-author network

We first integrate serendipity into co-author network $G = (V, E)$ for vector representation learning of author nodes. The node $v \in V$ in network represents the author, and edge $e \in E$ reflects the collaboration relationship between two connected nodes. We attach the serendipity of a collaborator for his/her target scholar to their edge weight. From Fig. 1, our co-author network is directed, because the edge weight of node $A$ to $B$ is different from that of $B$ to $A$. In other words, the serendipity brought by $A$ for $B$ is different from that of $B$ for $A$, since $A$'s relevance, unexpectedness and value scores for $B$ are different from that of $B$ for $A$.

Serendipitous collaborators are unexpected and valuable, but less relevant for their target scholars; therefore, we combine three indices as the edge weight in a linear way. The expression of edge weight is as follows:

$$w = \alpha \times RS + \beta \times US + \gamma \times VS, \tag{1}$$

where $RS, US$ and $VS$ represent the relevance score, unexpectedness score, and value score, respectively. While $\alpha$ is smaller than $\beta$ and $\gamma$, as $\alpha$ determines the proportion of relevance score, and $\alpha + \beta + \gamma = 1$. We aim to adjust these parameters and find the optimal collocation of them by analyzing the experimental results.

## Seren2vec

We improve the traditional DeepWalk model to learn vector representations of author nodes in co-author network for our recommendation task. In terms of DeepWalk, the walker during random walk jumps to the next node with the equal probability, $\frac{1}{N}$, where $N$ represents the number of last node's collaborators. DeepWalk excludes the importance or corresponding attributes of nodes. Take the random walk process in Fig. 1 as an example, the walker arrives at node $A$, and continues to walk to the next node. It will walk to $B$ with the probability of $\frac{1}{3}$ according to DeepWalk. In this work, we distinguish the importance of all collaborator nodes based on their serendipity extent for their target nodes. The extent of serendipity brought by collaborator $B$ for target scholar $A$ corresponds to their edge weight $w1$. If $w1$ is higher than $w2$ and $w3$, the walker will jump to $B$ from $A$ with higher probability.

We attach serendipity to the edge weight in co-author network, and finally guide to a serendipity-biased DeepWalk. As a consequence, the representation of the author vector will carry the attribute of serendipity. If a collaborator is serendipitous for a target scholar, his/her vector representation is similar with that of target scholar. The complete algorithm is described in Algorithm 1, where $b$ represents the set of betweenness centralities with respect to each author node, and $b_i$ denotes the betweenness centrality of

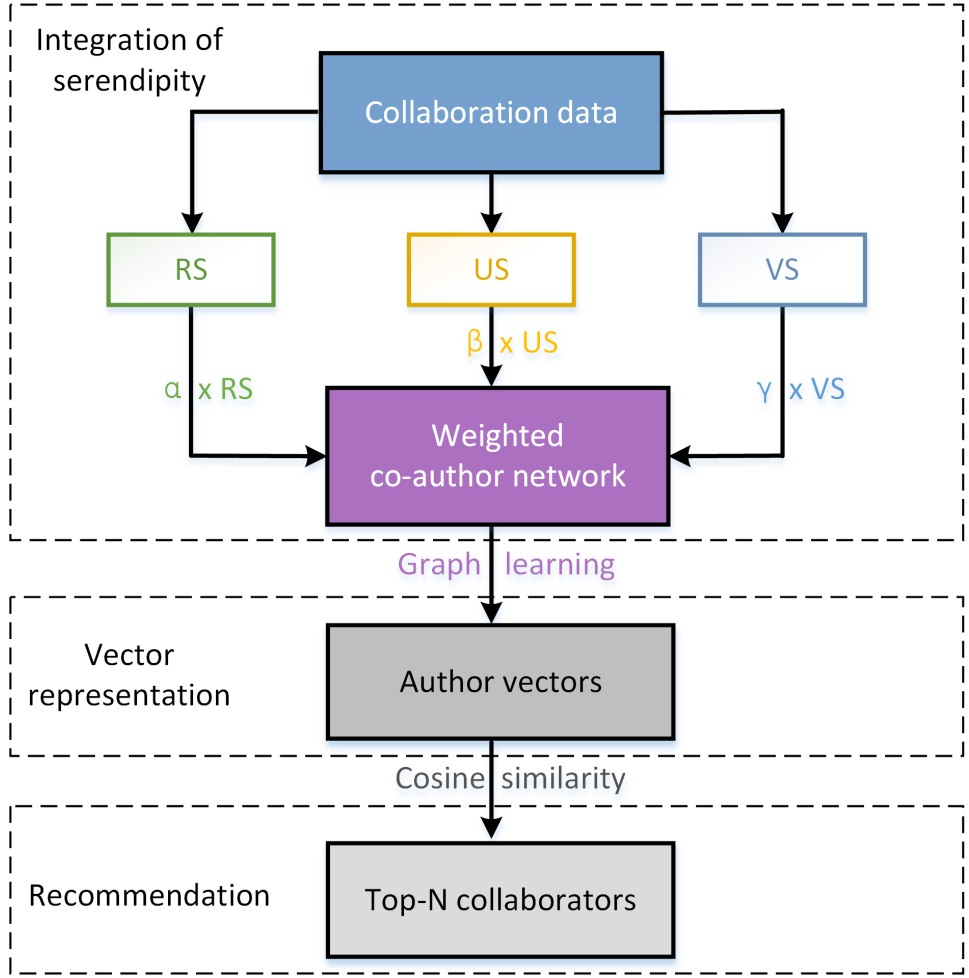

**Figure 2** **The flow chart of Seren2vec.** Seren2vec includes three main processes: integration of serendipity into DeepWalk, vector representation learning of each scholar, and collaborators recommendation based on the cosine similarity between vectors.

node $i$. $RS(i, j)$, $US(i, j)$, $VS(i, j)$ correspond with the relevance score, unexpectedness score, and value score of collaborator $j$ for his/her target scholar $i$, respectively. The flow chart of Seren2vec is shown in Fig. 2.

## Vector representation of Graph

The random walker walks on the co-author network with the probability of $p(i, j)$ from the scholar node $i$ to its collaborator node $j$:

$$p(i,j) = \frac{Weight(i,j)}{\sum_{q \in M(i), q \neq i} Weight(i,q)},$$ (2)

where $M(i)$ is the set of scholar $i$'s collaborators. We can find from *Perozzi, Al-Rfou & Skiena (2014)* that if the random walk algorithm is performed on the network with power law distribution, the nodes being visited also follow the power law distribution. This is the same distribution with the term frequency in natural language. Therefore, it is reasonable

---

**Algorithm 1:** Seren2vec

**Input:** Graph $G = (V, E)$, transfer matrix $T$, betweenness centrality set $b$, community degree set $c$, eigenvector centrality set $e$, parameter $\alpha$, parameter $\beta$, parameter $\gamma$, context size $w$, dimension of vertex vector $d$, walks per vertex $r$, walk length $l$, length of recommendation list $N$

**Output:** Recommendation List $Rec$

1    **for** $edge(i,j) \in E$ **do**
2      $\quad RS(i,j) = RWR(j, T)$;

3    **for** $edge(i,j) \in E$ **do**
4      $\quad US(i,j) = b_j + c_j$;

5    **for** $edge(i,j) \in E$ **do**
6      $\quad VS(i,j) = e_j$;

7    Build the weighted co-author network;

8    **for** $edge(i,j) \in E$ **do**
9      $\quad Weight(i,j) = \alpha \times RS(i,j) + \beta \times US(i,j) + \gamma \times VS(i,j)$;

10   **for** $k = 0; k < r; k{+}{+}$ **do**
11     **for** $u \in V$ **do**
12       $\quad walk[u] = Biased\ Random\ Walk(G, u, l)$;

13   $P = SkipGram(walk, w)$;
14   **for** $t = 0; t < row(P); t{+}{+}$ **do**
15     $\quad Rec[t] = Top\text{-}N\ similar\ collaborators$;

16   **return** $Rec$;

---

to regard a node in network as a word, and consider all the previous vertices being visited in the random walk as the sentence. Moreover, we utilize the Word2vec (*Mikolov et al., 2013*) model to input the sentence sequences generated during random walk into the Skip-Gram model, and take advantages of the stochastic gradient descent and the back propagation algorithm to optimize the vector representations. Finally, we obtain the optimal vector representation of each vertex in our co-author network.

Specifically, in order to learn the vector representations of vertices, Seren2vec makes use of the local information from the truncated random walks by maximizing the probability of a vertex $v_i$ in terms of the previous vertices being visited in the random walk:

$$P(\{v_{i-w}, \ldots, v_{i+w}\} \setminus v_i | \phi(v_i)) = \sum_{j=i-w, j \neq i}^{i+w} P(v_j | \phi(v_i)), \qquad (3)$$

where $\phi$ denotes the latent topological representation associated with each vertex $v_i$ in the graph, and $\phi$ is a matrix with $|V| \times d$. For speeding the training time, Hierarchical Softmax is used to approximate the probability distribution by allocating the vertices to the leaves

 

of the binary tree, and we compute $P(v_j|\phi(v_i))$ as follows:

$$P(v_j|\phi(v_i)) = \sum_{l=1}^{\lceil log|N|\rceil} \frac{1}{1+e^{-\phi(v_i)\cdot\psi(s_l)}}, \tag{4}$$

where $\psi(s_l)$ represents the parent of the tree node $s_l$. In addition, $(s_0, s_1, \ldots, s_{log|V|})$ is the sequence to detect the vertex $v_j$, and $s_0$ is the tree root.

The output of Seren2vec is the latent vector representations of all scholar nodes with $d$-dimension in our co-author network. We calculate the cosine similarity between other vectors with the target scholar vector $v_i$:

$$sim(x_{v_i}, x_{v_j}) = \frac{x_{v_i} \cdot x_{v_j}}{\sqrt{|x_{v_i}| \cdot |x_{v_j}|}}, v_j \in V \quad and \quad v_j \neq v_i. \tag{5}$$

Finally, Top-N similar scholars, who are serendipitous to the target scholar, are contained in the collaborator list for recommendation.

# EXPERIMENTS

In this section, we analyze and compare the performance of Seren2vec with other baseline methods for the serendipitous collaborators recommendation. We initialize the context size $w$ as 10, the vector dimension $d$ as 128, number of walks $r$ as 10, and walk length $l$ as 80 for conducting experiments.

## Data set

We extract collaboration data from five areas including Artificial Intelligence, Computer graphics and multimedia, Computer Networks, Data Mining and Software Engineering in DBLP data set. The co-author network is built by utilizing the collaboration data from year 2010 to 2012, and there are 49,317 nodes and 242,728 edges in the network. We randomly choose 100 authors who have one collaborator at least as the target scholars, and the final goal of our work is to recommend the serendipitous collaborators for them via Seren2vec.

## Baseline methods
### DeepWalk

DeepWalk (*Perozzi, Al-Rfou & Skiena, 2014*) is a widely used network embedding algorithm, which learns the latent vector representations of nodes in a network. It takes a graph as input and outputs the latent vector representations of all nodes in graph. DeepWalk can be easily utilized to obtain the author vectors. The core idea of DeepWalk is to take the random walk paths in network as sentences, and nodes as words. Applying the sentence sequences to SkipGram enable to learn the distributed representations of nodes.

### Node2vec

Node2vec (*Grover & Leskovec, 2016*) is a improved version of DeepWalk. It improves the strategy of random walk through the parameters $p$ and $q$, and considers the macrocosmic and microcosmic information simultaneously. Node2vec controls the transition probability of walker, where $p$ represents the returning probability, and $q$ denotes the probability of jumping to the next node. While DeepWalk sets both $p$ and $q$ at 1, and ignores other factors that will influence the generation of sentence sequences.

### *RWRW*

We also compare our Seren2vec with the baseline algorithm which does not adopt the vector representation strategy. Therefore, we run the serendipity-biased random walk with restart algorithm on our weighted co-author network (RWRW) for comparison.

### *TANGENT*

The TANGENT (*Onuma, Tong & Faloutsos, 2009*) algorithm is designed to recommend the relevant collaborators who have connectivity to other groups.

### *KFN*

The KFN (*Said et al., 2013*) algorithm aims to recommend the novel collaborators who are disliked by the dissimilar neighborhood of the target scholar. This approach is contrary to K nearest neighbors (KNN) (*Deng et al., 2016*).

## Evaluation metrics

We refer to the metrics in *Ge, Delgado-Battenfeld & Jannach (2010)*, *Zheng, Chan & Ip (2015)* and *Kotkov, Wang & Veijalainen (2016)* for evaluating our serendipitous collaborators recommendation, which are the serendipity-based metrics including unexpectedness, value and serendipity. We compute the average RS and US of each target scholars's collaborators. If RS of one collaborator is higher than the average RS, and on the contrary, his/her US is lower than the average US, we regard this collaborator as the expected one of target scholar. Expected collaborators are more relevant and less unexpected than other collaborators for their target scholars. According to *Ge, Delgado-Battenfeld & Jannach (2010)* and *Zheng, Chan & Ip (2015)*, a recommended collaborator not in the set of target scholar's expected collaborators is considered as unexpected. Therefore, we extract the expected collaborators of each target scholar first, and a recommended collaborator not in the expected set is evaluated as unexpected. The unexpectedness is measured as the rate of unexpected collaborators in the recommendation list.

In terms of the metric of value, *Zheng, Chan & Ip (2015)* measured the usefulness of a recommended item through its rating given by the target user. While in our recommendation scenario, we analyze the collaboration times between the recommended collaborator and target scholar in the next 5 years. If their collaboration times is over or equal to 3 times from 2013 to 2017, the recommended collaborator is evaluated as valuable. Besides, the value is measured as the rate of valuable collaborators in the recommendation list.

*Ge, Delgado-Battenfeld & Jannach (2010)* combined the unexpectedness and value metric to evaluate serendipity. Similarly, we measure serendipity through the rate of collaborators who are both unexpected and valuable for the target scholar in the recommendation list.

## Recommendation results analysis

We take the widely used serendipity-based metrics to evaluate our recommendation results, including unexpectedness, value and serendipity. Therefore, the evaluations are no longer limited by the accuracy-based metrics. In this section, we examine the effects of different experimental parameter sets on the recommendation performance, including the range of $d, r, l, w$, different combinations of $\alpha, \beta$ and $\gamma$, and the length of recommendation list $N$.

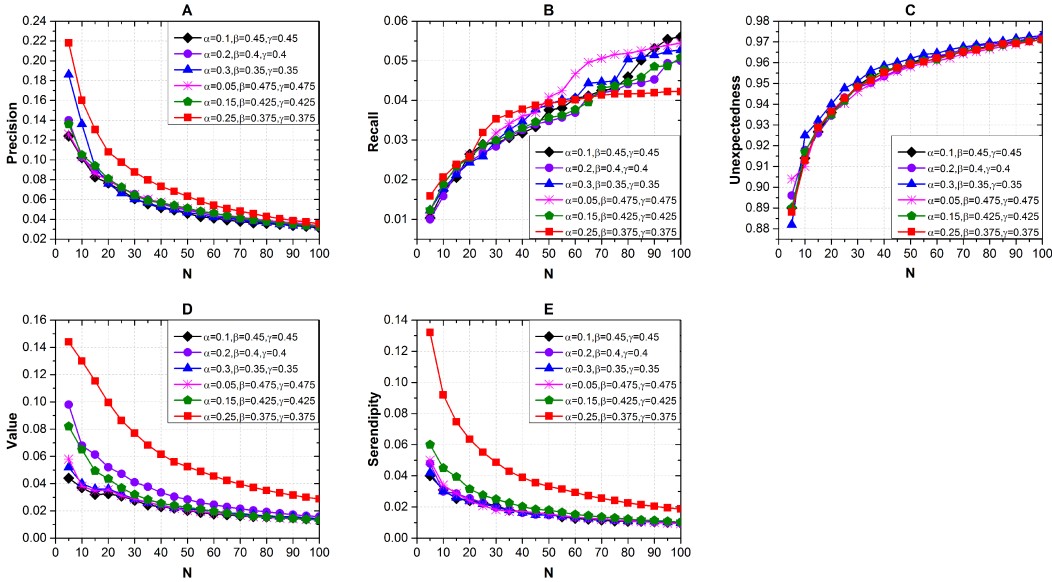

**Figure 3** **Effects of the recommendation list length and different combinations of $\alpha$, $\beta$ and $\gamma$ on recommendation performance via Seren2vec.** (A) Precision (B) Recall (C) Unexpectedness (D) Value and (E) Serendipity of the recommendations. (When $\alpha = 0.25$, $\beta = 0.375$, and $\gamma = 0.375$, Seren2vec shows the highest precision, value and serendipity. Meanwhile, the optimal $N$ is 5).

### Effects of the recommendation list

We measured the recommendation results with different length of recommendation list $N$ and different combinations of $\alpha, \beta$ and $\gamma$ via Seren2vec. The results shown in Fig. 3 indicate that when $\alpha = 0.25$, $\beta = 0.375$ and $\gamma = 0.375$, the recommendation performance is better than others with respect to the precision Fig. 3A), value (Fig. 3D) and serendipity (Fig. 3E) evaluations. We also find that the optimal $N$ is 5. The precision, value and serendipity decrease with the increases of $N$, which are contrary to the recall (Fig. 3B) and unexpectedness (Fig. 3C). Furthermore, the unexpectedness of different combinations keeps close distributions with the variation of $N$.

Furthermore, we show the effects of the recommendation list length with $\alpha = 0.25, \beta = 0.375$, and $\gamma = 0.375$ via different baseline methods. From Fig. 4, Node2vec obtains the highest precision (Fig. 4A) and recall (Fig. 4B), but it shows the worst performance from the serendipity-based metrics (Figs. 4C–4E) evaluation. Our Seren2vec outperforms others in terms of the serendipity-based metrics, and keeps adequate accuracy simultaneously. RWRW has lower precision than Seren2vec because it lacks the vector representation process, but it has the second highest serendipity finally because of the serendipity-biased random walk. Meanwhile, the collaborators recommended by Seren2vec show higher unexpectedness than others when the length of recommendation list is lower than 60. Node2vec and Deepwalk share close serendipity with low value, because they have not integrated other serendipity-related elements into the vector representation learning.

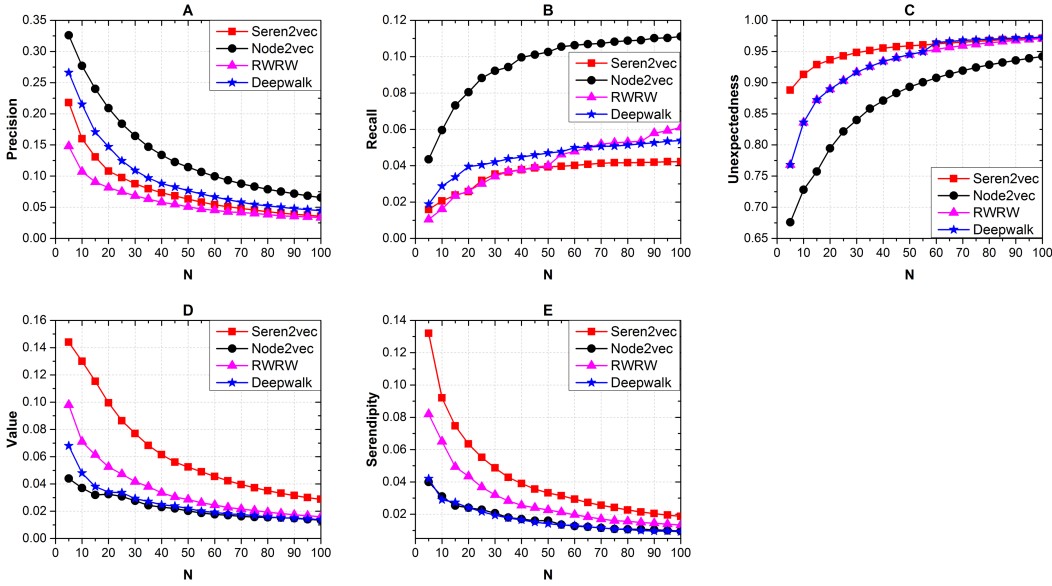

**Figure 4** **Effects of the recommendation list length with $\alpha = 0.25$, $\beta = 0.375$, and $\gamma = 0.375$ on recommendation performance via different baseline methods.** (A) Precision (B) Recall (C) Unexpectedness (D) Value and (E) Serendipity of the recommendations. (Node2vec shows the highest accuracy. Seren2vec shows the highest serendipity, and it is superior to RWRW in terms of the precision evaluation).

### Parameter sensitivity

We tested the recommendation performance with different parameters, and the length of recommendation list is optimally five according to the last subsection. When testing one parameter of $l$ (Fig. 5A), $r$ (Fig. 5B), $d$ (Fig. 5C), $w$ (Fig. 5D) with different values on the recommendation performance, we fix other three parameters with corresponding initial values. We can get from Fig. 5 that the measurements from both accuracy-based and serendipity-based metrics almost have the steady distributions under different parameter sets. We take the set where $r = 80$, $d = 208$, $l = 80$ and $w = 6$ as our final parameter set, since each of them contributes to the highest serendipity and maintains adequate accuracy simultaneously.

### Performance comparison

Next, we compare our Seren2vec which are assigned to the optimal parameters with other two baseline methods. We set the length of recommendation list of TANGENT and KFN as 5, which is the same with Seren2vec. From the comparison results in Fig. 6, Seren2vec obtains better performance for recommending the serendipitous collaborators via the evaluations from serendipity-based metrics, since it adopts the serendipity-biased representation learning strategy. Its highest value contributes to the highest serendipity. TANGENT and KFN stress the unexpectedness of recommendations, but they ignore the value component of serendipity. Therefore, they are inferior to Seren2vec in terms of the serendipity evaluation. Seren2vec outperforms other two methods except for the unexpectedness evaluation, which is 0.014 lower than the unexpectedness of TANGENT,

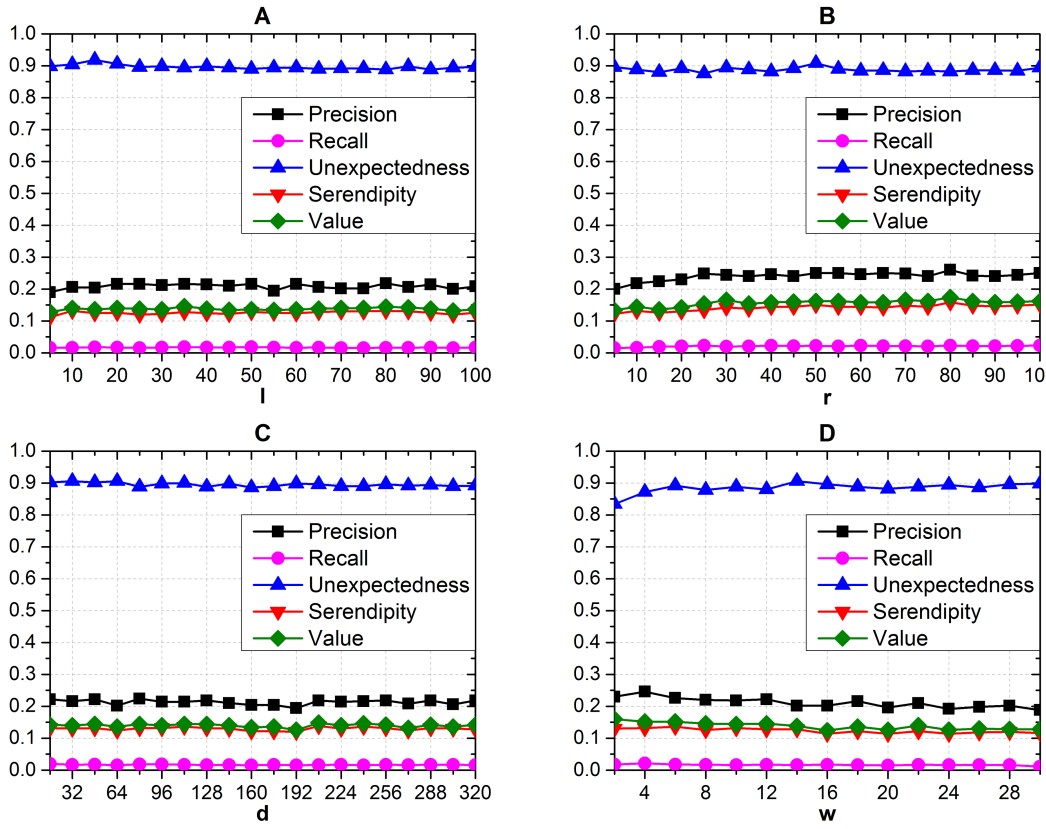

**Figure 5  Effects of different *d*, *r*, *l* and *w* on recommendation performance.** (A) The effects of *l* (B) *r* (C) *d* and (D) *w*. (Seren2vec almost keeps steady distribution on recommendation performance with the variation of parameters).

but 0.048 higher than that of KFN. Furthermore, we find that the recall of three methods are very low, and Seren2vec has higher recall than TANGENT and KFN slightly.

In summary, our proposed Seren2vec is superior to other baseline methods for recommending more serendipitous collaborators. It learns the serendipity-biased vector representation of each author node in co-author network successfully, which integrates the serendipity between two connected nodes.

## CONCLUSION

This paper introduces the scientific collaborators from a new perspective of serendipity. The serendipitous collaborator has high topic diversity, high influence and low proximity in co-author network for his/her target scholar relatively. We focused on designing an effective algorithm to integrate serendipity into collaborators recommender system. Specifically, we improved the DeepWalk algorithm, where the sentence sequences are generated by performing a serendipity-biased random walk. Seren2vec represents the author nodes in the co-author network with the low-dimensional vectors, which are attached with the attributes of serendipity successfully. Finally, we computed the cosine similarity between

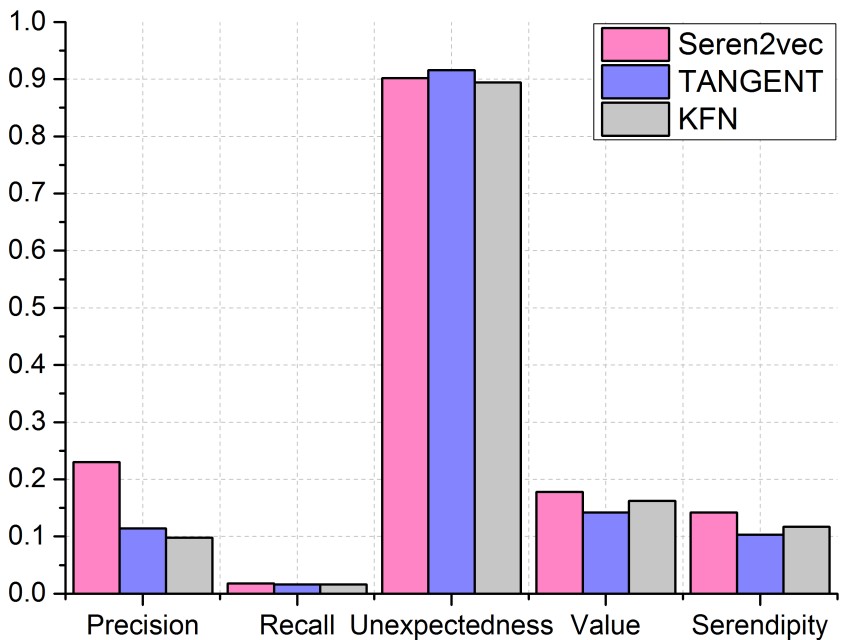

**Figure 6** **Performance comparisons between different recommendation methods.** (Seren2vec is superior to TANGENT and KFN by evaluating from both precision and serendipity metrics).

the vector of target scholar and other vectors, and extracted Top-N similar collaborators for recommendation. The extensive experiments are conducted on the DBLP data set, and the experimental results show that Seren2vec is more effective than other baseline approaches by evaluating from the serendipity-based metrics. Seren2vec improves the serendipity of recommendation list and maintains adequate accuracy simultaneously.

### Funding
The authors received no funding for this work.

### Competing Interests
The authors declare there are no competing interests.

### Author Contributions
- Zhenzhen Xu analyzed the data.
- Yuyuan Yuan conceived and designed the experiments, prepared figures and/or tables, performed the computation work, authored or reviewed drafts of the paper.
- Haoran Wei performed the experiments, performed the computation work.
- Liangtian Wan contributed reagents/materials/analysis tools, authored or reviewed drafts of the paper, approved the final draft.

## Data Availability

Data is available at https://snap.stanford.edu/data/com-DBLP.html.

## Supplemental Information

Supplemental information for this article can be found online at http://dx.doi.org/10.7717/peerj-cs.178#supplemental-information.

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
