# Peer review of "A serendipity-biased Deepwalk for collaborators recommendation"

_PeerJ Computer Science, doi:10.7717/peerj-cs.178_

## Round 0.1 · original submission · Major Revisions

Please improve the language of the paper. In addition, please improve the quality of the provided figures. Finally, please include a more complete review on why your work relates with the literature (see e.g. papers on time-varying collaboration networks).


Reviewer 1 ·

Basic reporting

Although there are some works in serendipitous recommendation, the problem of recommending serendipitous scientific collaborators is interesting. The innovation and novelty of this research are the definition and strategy of serendipity-biased network embedding. Their method balances the serendipity and accuracy of recommendation list effectively.

The paper proposes a method for identifying serendipitous collaborators. It defines serendipity in co-author network based on relevance, unexpectedness and value. These indices are well defined and works together to define serendipity. They improve the DeepWalk to learn the serendipity-biased vector representation of each vertex based on the linear combination of three indices. The proportion of relevance is smaller than unexpectedness and value according to their definition. I suggest authors to describe how to divide the research areas of scholars by LDA in the subsection of unexpectedness score.

Experimental design

To evaluate their work, they not only adopt the existing accuracy-based metrics, but also refer to the serendipity-based metrics in literature to measure their recommendation results. Their method outperforms other baseline methods in terms of the serendipity measurement, and maintains adequate accuracy at the same time.

Validity of the findings

The research is well motivated. The authors aim to design method for overcoming the defects of the traditional recommender system. The paper is also heuristic and applicable in scientific collaboration, and it may guide and inspire future efforts on the research of serendipity.

Additional comments

Minor comments:
1. Line 218 in page 5: value score-> value scores.
2. Line 384 in page 11: by evaluating the serendipity-based metrics -> by evaluating from the serendipity-based metrics.

Reviewer 2 ·

Basic reporting

Please see 'General comments for the author'

Experimental design

Please see General comments for the author

Validity of the findings

Please see General comments for the author

Additional comments

The topic of this paper is fresh and interesting as it explores the concept of serendipitous scientific collaborators and designs a serendipity-biased recommender system. The components of relevance, unexpectedness and value are well quantified to define the serendipitous collaborators in the context of collaborators recommendation. The authors combine three quantified scores linearly to guide a serendipity-biased DeepWalk for leaning the vector representation of author node.

This research is novel and innovative, it tackles a fundamental problem in recommendation: the trade off between serendipity and accuracy. The scientific production scenario is perfect to analyze this problem given that diversity is a key point in the creativity process. I can see the expected characteristics of a serendipitous collaborator is: connection in the research network with the scholar target, influence in the research network and diversity in topic researches.

The paper introduces the motivation, goals and contributions of the presented research in the first section. The analysis of the literature in related work are clear and define well the scope of the paper. The authors take some recommendation methods from the literature cited as baseline methods for comparison, and refer to some serendipity-based metrics for measuring their recommendation results.

Their framework is improved from the traditional DeepWalk, where the sentence sequences are generated by performing a serendipity-biased random walk. The walker jumps to the next neighbor node with the proportional probability of edge weight in co-author network, where the edge weight is determined by the linear combination of three scores in definition. Finally, they compute the cosine similarity between vectors and extract the Top-N scholars who are serendipitous for target scholars. The experimental results validate the effectiveness of their method via the performance comparison between different baseline methods. Multiple parameter sets are also tested and analyzed in experiments for selecting the optimal set.

The paper reports a complete cycle from motivation to concept and technique development, and experimental observation.

Minor comments:
1. Line 200 in page 4: it across  it crosses.
2. Line 221 in page 5: relevance, unexpectedness and value score  relevance, unexpectedness and value scores.
3. Line 384 in page 11: by evaluating the serendipity-based metrics  by evaluating from the serendipity-based metrics.

---

## Round 0.2 · Minor Revisions

Please follow all referees' recommendations, and especially those regarding text and figure quality.

Reviewer 1 ·

Basic reporting

no comment

Experimental design

no comment

Validity of the findings

no comment

Additional comments

All my concerns have been issued well.

Reviewer 2 ·

Basic reporting

no comment.

Experimental design

no comment

Validity of the findings

no comment

Additional comments

The revisions are satisfactory.

Reviewer 3 ·

Basic reporting

All the reviewers' suggestions were addressed and the manuscript was improved for what concerns the clarity of the presented methodology and results, as well as, the text correctedness.

Experimental design

I have just one minor request. The plots of figures 3, 4 and 5 should be improved regarding its arrangement in the manuscript. Also, the plots of theses figures should be larger. I think it is better, for instance, to split the plots of precision, recall, and so on, in more rows per figure. This way the readability of those figures will be improved. The way they are shown now, the reader struggles to visualize the information. The plot of Figure 6 should also be improved.

Validity of the findings

The main topic of the paper, the serendipitous scientific collaborators, is very interesting and is well approached by the authors. The experimental setup is effective and the evaluation methods are statistically sound in order to validate the obtained results.

Reviewer 4 ·

Basic reporting

- Professional English, clear, and coherent. Only small corrections need to be made in some text segments at lines 44 and 46.
- It would be important to review the abstract of the article again.

Experimental design

no comment

Validity of the findings

no comment

Additional comments

- The proposal of the article in approaching the problem of scientific collaboration recommendation is interesting, innovative and novel. The authors proposed the implementation of a recommendation system to generate ideal or appropriate scientific collaborators among scholars. In addition, representative vectors or embeddings were generated for each author or scholar (each node of a co-authoring network) proposing the Seren2vec methodology, which was compared with different algorithms for vector representation of nodes such as Node2vec, DeepWalk, among others. A series of experiments were performed with a significant and important database such as DBLP. In addition, these experiments were evaluated using different existing metrics in the literature for the validation of recommendation systems. The results showed that the proposed method surpasses other baseline methods that also maintain a correct accuracy.
- The article does not have many things to correct, because it is written with a clear, coherent and professional English. In addition, the article is well structured, organized and the proposal and results are clearly defined. The text segments that need to be corrected are the following:
* Line 44: quantified by us as -> quantified as
* Line 46: the text “research topics from that of their target scholars” is not clear, it needs to be improved.
- The analysis of the literature is quite good, well ordered and structured since it clearly divides the main methods that address the task of recommendation systems. It is also good that some works about serendipity in recommender systems are presented.
- Regarding the abstract, with the aim of making it clearer, it would be convenient to make a small modification regarding the use and definition of the term Serendipity, since it is a little-known term. At the beginning you name several terms that include the word Serendipity (serendipitous scientific collaborators, serendipity-biased vector representation or serendipitous encounters) and after you give an explanation about that term. It would be ideal that you first try to define the term Serendipity with some related examples and then you use the terms that include the word Serendipity (serendipitous scientific collaborators, serendipity-biased vector representation or serendipitous encounters).

---

## Round 0.3 · accepted · Accept

=========================

Reviewer 3 ·

Basic reporting

no comments

Experimental design

no comments

Validity of the findings

no comments

Additional comments

All the requested changes were made by the authors.

Reviewer 4 ·

Basic reporting

no comment

Experimental design

no comment

Validity of the findings

no comment

Additional comments

The revisions are satisfactory.